# Test-retest reliability and validity of the Importance of Olfaction Questionnaire in Denmark

Daniel Tchemerinsky Konieczny[1]*, Alexander Wieck Fjaeldstad[2,3,4], Kristian Sandberg[1,5]

1 Center of Functionally Integrative Neuroscience, Aarhus University, Aarhus C, Denmark, 2 Flavour Institute, Aarhus University, Aarhus N, Denmark, 3 Department of Otorhinolaryngology, Flavour Clinic, University Clinic for Flavour, Balance and Sleep, Regional Hospital Gødstrup, Central Denmark Region, Herning, Denmark, 4 Center for Eudaimonia and Human Flourishing, Linacre College, University of Oxford, Oxford, United Kingdom, 5 Center of Functionally Integrative Neuroscience, Aarhus University Hospital, Aarhus N, Denmark

☯ These authors contributed equally to this work.
* danieltchemerinskij@gmail.com

## Abstract

While measures to detect psychophysical olfactory ability are a crucial part of clinicians' assessment of potential olfactory loss, it gives no indication of how olfaction is experienced by the patient and these different aspects often deviate substantially. To ensure quality and reproducibility of subjectively reported olfactory experience and significance, the Importance of Olfaction Questionnaire (IO-Q) was introduced around a decade ago, and while initial validations have produced promising results, important aspects remain nearly unexamined. For example, the test-retest reliability has rarely been examined and the difference of online versus pen-and-paper administration remains unexplored. Here, we translated IO-Q to Danish and examined its validity, test-retest reliability and mode of administration. A cohort of 179 younger, Danish participants with a high level of English proficiency took the test twice with varying time in-between. The first test was taken digitally and in English, while the second was taken using pen-and-paper and in Danish. The distribution of scores and the relationship between the IO-Q and subscale scores were nearly identical between tests, indicating little to no influence of language/test modality in the sampled population. The internal consistency was comparable to previously published results. Likewise, an acceptable test-retest reliability was observed for the full IO-Q and slightly lower for subscales. No significant effect of time was found across several weeks. In conclusion, the IO-Q performed satisfactorily in all examinations and could therefore serve as a valuable clinical measure of subjective olfactory experience, and its Danish translation shows highly similar characteristics to the original, English version.

## Introduction

For many otorhinolaryngological- and neurological disorders, it is crucial to have readily available and reliable clinical tools to assess olfactory functioning in the patients' native language,

**Data Availability Statement:** Data cannot be shared publicly because it is part of an ongoing study and thus considered unanonymised under Danish law even if pseudonymised. Researchers who wish to access the data may contact Dr Kristian Sandberg (kristian.sandberg@cfin.au.dk) at The Center of Functionally Integrative Neuroscience and/or The Technology Transfer Office (TTO@au.dk) at Aarhus University, Denmark, to make a data sharing contract. After permission has been given by the relevant data committee, data will be made available to the researchers.

**Funding:** This article is based upon work from COST Action CA18106 The Neural Architecture of Consciousness supported by the European Cooperation in Science and Technology (COST). Author AWF was supported by research funding from Hans Skouby Foundation and VELUX Fonden (grant number 00035282), and author KS by the National Science Centre, Poland, OPUS grant 2017/27/B/HS6/00937 and the Danish Foundation for Research in Neurology. The funders had no role in the study design, data collection and analysis, decision to publish, or preparation of the manuscript.

**Competing interests:** The authors have declared that no competing interests exist.

especially if the patient population has limited proficiency in their second language. Olfactory deficits are common, affecting up to 20% of the population [1]. In clinical and research settings, olfactory function is assessed by psychophysical testing of different aspects of olfactory function, including identification, discrimination and threshold.

The experienced distress and perceived loss reported in hyposmic and anosmic patients, however, varies greatly from person to person [2, 3]. Some will have nearly no complaints, while others report that the olfactory impairment causes significant impairment in daily functioning. Cooking and eating, for instance, can become bland and unfulfilling, while severely affecting dietary choices and the hedonic yield [4]. Likewise, in a normosmic population, the relationship between TDI scores and the perceived importance of olfaction is weak [5]. Both these findings indicate that the subjective importance of olfaction is often quite distinct from measured olfactory function and that it is thus important to measure it separately. A commonly applied test to examine this aspect is the Importance of Olfaction Questionnaire (IO-Q) [6]. It consists of 20 items with four distinct subscales, and it provides nuanced information on the individual parts of the olfactory experien2ce.

Previously, the subscales of the IO-Q have been found to be significantly correlated to each other and to the main score itself, and the internal reliability of the IO-Q is acceptable (Cronbach's alpha = 0.77) [6]. Currently, most results point to no significant coherence between age and overall IO-Q in normosmic populations [6, 7], although olfactory function decreases especially beyond 60 and 70 for men and women, respectively [8]. One study did, however, find a significant effect of age group on overall IO-Q in a normosmic population [9]. In another study, women tended to score higher in the subscale 'consequence' [7] with a later study finding a similar pattern in the subscale 'application' [9]. As such, the demographic dynamics and normative scores are still not fully characterized.

The current published data on test-retest reliability and validity of the IO-Q is sparse. The study presented here serves to further determine the test-retest reliability as well as the validity of the IO-Q in a healthy, primarily younger population in Denmark. As the availability of native-language testing is important, particularly in clinical samples, a main aim of the study was to translate the IO-Q into Danish and validate the translated version.

## Methods

All data was collected within the context of European Cooperation in Science and technology (EU COST) Action CA18106 and constitutes a part of a larger array of magnetic resonance imaging (MRI) and behavioral data collected from healthy, younger, and neurotypical individuals. The pen-and-paper IO-Q data presented here is used in another publication as a separate article with a different topic [5]. The regional ethics committee *De Videnskabsetiske Komitéer for Region Midtjylland*, Denmark, approved the study (Project ID: M-2016-69-16).

### Participants

The study examined healthy, young, Danish participants. Inclusion criteria were age between 18 and 50 years, physically healthy, not currently in medical treatment for psychiatric disorders, normal or corrected-to-normal vision, normal hearing and no major brain abnormalities (no known abnormalities, brain damage or brain surgery). Exclusion criteria were: MRI contraindications (e.g., non-MRI-safe metals in the body), use of medicine that might affect neural states, body build that does not allow MRI scanning and pregnancy. Participants were recruited from the Aarhus University Center of Functionally Integrative Neuroscience participant database as well as through advertisement in online social media student groups and by physical advertisement on campus and in dorms.

In total, 300 participants provided written informed consent and were included in the large-scale CA18106 study at Aarhus University, Denmark. They were recruited through the Center of Functionally Integrative Neuroscience (Aarhus University) participant database and through local advertisement, and they were compensated 120DKK per hour of participation. After exclusions (detailed in *Methods*: *Procedure*), 179 participants were included in the analyses. The median age was 27 (range 21 to 51 years of age). Participants were required to confirm good English skills and be comfortable with experimental testing in English as a perquisite to volunteer for the study.

The sample used within this study was fairly representative of the younger population in Denmark, although the participants were, on average, slightly more educated and had a higher intelligence quotient (IQ). Most of the sample was currently enrolled as a student on a formal education beyond secondary school (~81%) or had completed such an education. For comparison, 85% of Danes between ages 25 and 44 years have completed education beyond secondary school, and 40–45% have completed higher education of at least 3 years' duration [10]. The majority (92%) of the included participants completed a Wechsler Adult Intelligence Scale IV (WAIS-IV) test, and their average IQ was 112.7, thus being somewhat above the average of the population, which–in Denmark–is estimated to be 98 [11].

## Importance of olfaction

The total score of the IO-Q is composed of 20 items divided into four subscales (Association (Ass), Application (App), Consequence (Con), and Aggravation (Agg)). The three subscales Ass, App, and Con each consist of 6 items and Agg of 2 items. The first, Ass, refers primarily to which associations the sensation of smell might evoke such as emotions and memories (e.g., whether smelling delicious food makes one hungry); The second, App, refers to the goal-directed usage of olfaction (e.g., smelling food to find out if it is spoiled); The third, Con, refers to how heavily olfaction can influence decision-making (e.g., if one avoids buying a particular shampoo if they do not like the smell). Finally, Agg assesses how important olfaction is to life in general (e.g., in comparison to other senses), and it has been proposed as a gauge of overestimation in the participant [6]. On a 4-point Likert scale ranging from 'Totally Agree' to 'Totally Disagree', where the former is scored as 3 and the latter as 0, each item is tallied and totaled. As such, each subscale has a total maximum score of 18 (6 for Agg) with a total maximum score of 60. A greater score indicates greater subjective olfactory importance.

Native Danish speakers with bachelor's degrees in the English language translated the IO-Q. Two translators translated from English to Danish, and one translated back to English. The original and back-translated versions were compared and verified by author AWF. All four then compared the two translated Danish versions and reached a consensus on a final version (S1 Table).

## Procedure

The questionnaire was first administered in an online (digital) format in English, and then later–with a variable inter-test duration–in a pen-and-paper (analogue) format in Danish. The first test was administered online for practical purposes as it was part of a fully online session that participants completed from home. The second test was administered as pen-and-paper to keep it as similar to clinical testing as possible. The duration between tests was determined by the time between the two different experimental sessions, which participants could book freely. Participants were instructed to always complete the online session with the digital questionnaire before booking the later session with the analogue version. One participant did not follow this instruction and was excluded from the study. A total of 213 participants completed

both tests and of these, 183 provided a full Danish IO-Q response without missing or invalid answers. Three participants had a very long and not representative duration of ~45 or more weeks between sessions and were excluded. For the remaining 179 participants, the duration between tests had a median of 2.3 weeks (range: 0.3–18.7 weeks).

## Statistics

Statistical analyses were performed using Bayesian tests. The Bayes Factor (BF) represents the odds of either the alternate hypothesis (BF10) or the null hypothesis being true (BF01). This stands in contrast to the traditional, frequentist approach in which it cannot be determined how likely the alternate hypothesis is given the data observed. A BF of 1–3 is often categorized as constituting anecdotal evidence, 3–10 substantial evidence, 10–30 strong evidence, 30–100 as very strong evidence, and >100 extreme evidence [12]. When using previous results as priors, log(BF) is reported. Note that for these log(BF) > 4.6 constitutes extreme evidence, since the natural logarithm of 100 is 4.61 For all correlations except replicative analysis (which use test 1 as a prior), we used a non-directional uniform prior of -1 to 1, which consequently sustains compatibility to other studies using parametric frequentist correlation [13]. In addition, a Bayesian correlational analysis also produces a posterior distribution constituting the best estimate of the effect size given the prior and the data, with a peak denoted the maximum a posteriori probability (MAP), the correlation reported, and a 95% credible or confidence interval.

To maintain comparability with previous studies, parametric tests are reported in the results section. As detailed later in this section, some (often minor) deviations from the test assumptions were observed and for transparency, we report non-parametric analyses in the Appendix. Pearson product moment correlations were used to examine the test-retest reliability of the IO-Q. For convergent validity analyses, the four subscales of the IO-Q were correlated among each other and to the composite total score using Pearson product moment correlations with an added measure of Cronbach's alpha. To include the a priori information gathered during the first round of testing, we also performed a Bayesian correlation analysis using the first round of testing as the prior. In the same vein, the first and second test round were always used as a prior for comparison to results previously published by Croy et al. (2010).

Further, Bayesian t-test was used to test for evidence of the alternative hypothesis that there is a difference between the two tests. Finally, to understand the probable effect of time on the scaled difference between test scores, a graphical posterior predictive check was made. It is essentially an iterative simulation of 10,000 runs to see if a fitted model can predict the distribution of the data and therefore could be considered an estimate of the goodness of fit. The analysis was conducted using the open-source environment R version 4.1.1. mainly with the package collection 'tidyverse' [14] as well as JASP [15]. Priors in the replication correlation analysis were set by using the data from the first testing round to obtain a posterior distribution, which was then applied as the alternative hypothesis in the analysis. The method is previously published [16, 17], and R code is made freely available by one of the authors (Verhagen) (currently available at https://www.josineverhagen.com/?page_id=76). We used this code as a template for our analysis.

Apart from sex, each variable examined in this study was continuous, and each observation had a pair of values (i.e., first and second time of testing). Considering the sample of 179 analyzed participants, there were few outliers, particularly with respect to time between tests. Linearity and homoscedasticity using the Breusch-Pagan test with visual inspection did not reject that the variance of the residuals was constant. However, some of the subscales did not pass all assumptions; particularly common were minor deviations from a normal distribution.

## Results

### Means, distributions, and internal consistency

The total IO-Q score distributions for all participants for Test 1 (English) and Test 2 (Danish) are shown in Fig 1 in raincloud plots. Only Agg failed to exhibit an approximately normally

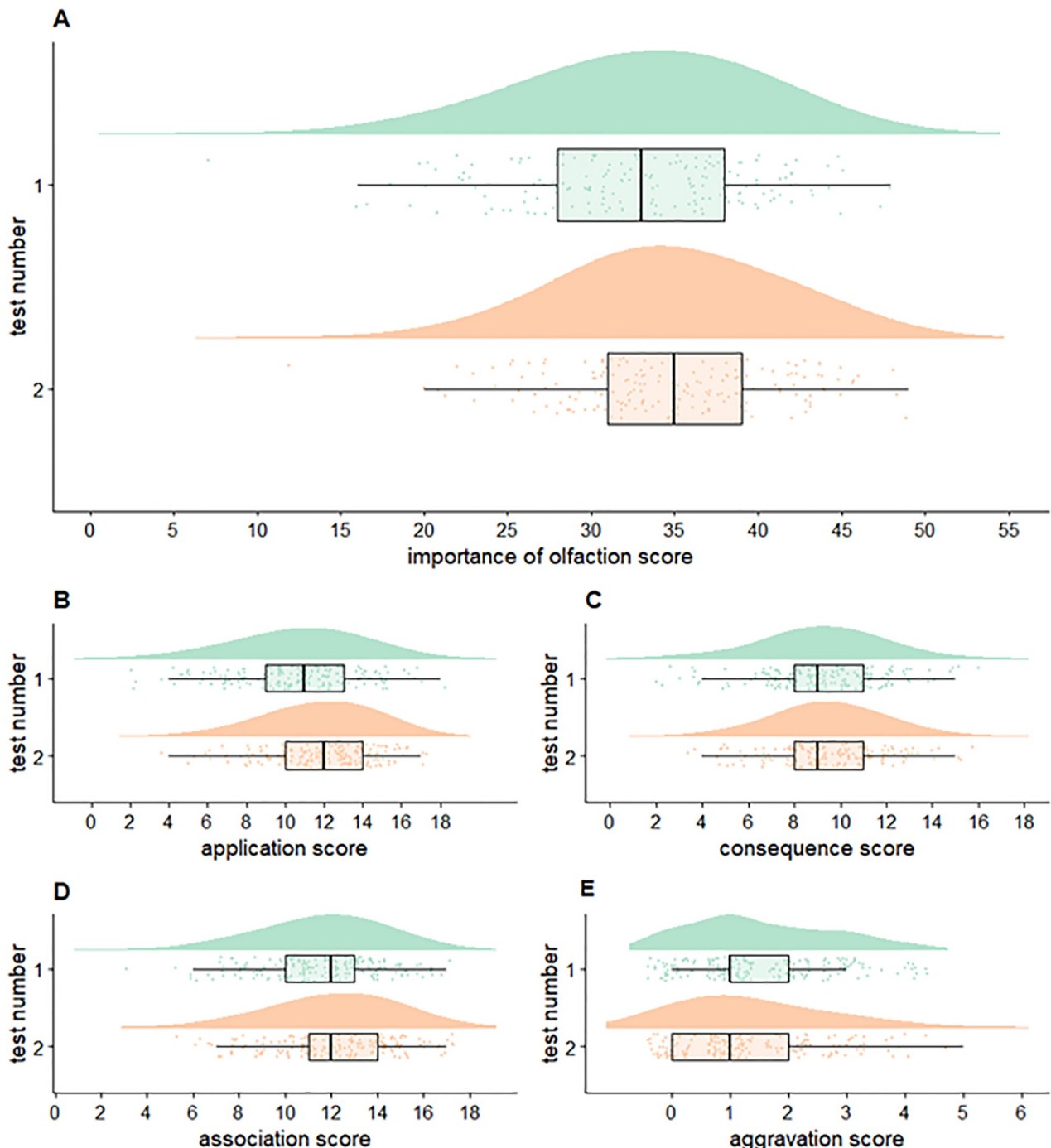

**Fig 1. Distributions of Importance of Olfaction Questionnaire (IO-Q) scores.** Raincloud plots [22] of the score distributions for total IO-Q (A) and subscales (B-E) for tests 1 and 2. The 'cloud' illustrates the data distribution, while below, observations are marked as jittered dots. On the dots, a boxplot is placed.

**Table 1.  Descriptive table of all scores.**

|  | n | Mean (SD) | Median (Q1; Q3) |
|---|---|---|---|
| Age | 179 | 27.94 (5.31) | 27 (25; 29) |
| Females | 100 (~56%) |  |  |
| First IO-Q | 179 | 32.93 (6.78) | 33 (28; 38) |
| Second IO-Q | 179 | 34.71 (6.25) | 35 (31; 39) |
| First association | 179 | 11.59 (2.59) | 12 (10; 13) |
| Second association | 179 | 12.14 (2.58) | 12 (11; 14) |
| First application | 179 | 10.68 (2.58) | 11 (9; 13) |
| Second application | 179 | 11.78 (2.66) | 12 (10; 14) |
| First consequence | 179 | 9.12 (2.57) | 9 (8; 11) |
| Second consequence | 179 | 9.46 (2.25) | 9 (8; 11) |
| First aggravation | 179 | 1.54 (1.18) | 1 (1; 2) |
| Second aggravation | 179 | 1.33 (1.15) | 1 (0;2) |

'First' signifies first testing round, and 'second' signifies second testing round. IO-Q is the total Importance of Olfaction score.

distributed pattern. Table 1 contains summary statistics for all scores. The internal consistency of the IO-Q is acceptable. (Cronbach's alpha = 0.733 (95% CI = [0.622;0.787]) for Test 1 and 0.746 (95% CI = [0.684; 0.795]) for Test 2) (Fig 2A).

Repeated testing might cause a general shift (increase or decrease) in numerical test scores even if the overall patterns/correlations stay constant. For this reason, we first examined the change in reports between tests 1 and 2, and we compared both to previously published results. Fig 2B–2F plots the means and standard deviations for the total IO-Q and all subscales in comparison to the results reported by Croy et al. in 2011 in a normosmic population of 235 participants (mean age = 27.2). Overall, scores were similar between tests 1 and 2. Nevertheless, the total IO-Q score was on average 1.8 points higher on the second test and Bayesian Students paired sample t-tests revealed extreme evidence for this difference (Table 2). The difference was largely driven by App and Ass being rated respectively 1 and 0.5 points higher (Table 2). It may be noted that 1.8 is numerically a relatively small difference despite the clear statistical evidence of a difference.

Comparing our results to those of Croy and colleagues from 2011, numerical differences ranging up to around 1.5 were found for the individual subscales (Table 3). Numerically, the largest differences were found for Con, and statistically, the clearest evidence was for Agg. In both these cases, our participants rated lower on both the first and the second tests. For Ass, nearly no difference was found, and this was also the case for the first App test. On the second App test, our participants rated higher than those of Croy and colleagues. Taken together, some minor average differences were thus found both when comparing the same participants over time and when comparing between samples. As the differences were relatively small and similar in magnitude across samples as over time within the sample, it is likely that the differences reflect scale reliability more than meaningful participant differences. As an addendum to this comparative analysis, we include a table with results from additional studies in S2 Table.

## Correlation between subscales and total IO-Q

The internal correlations between subscales/total IO-Q is shown for tests 1 and 2 separately in two correlation matrices (Fig 3A and 3C). The correlation between total IO-Q and each

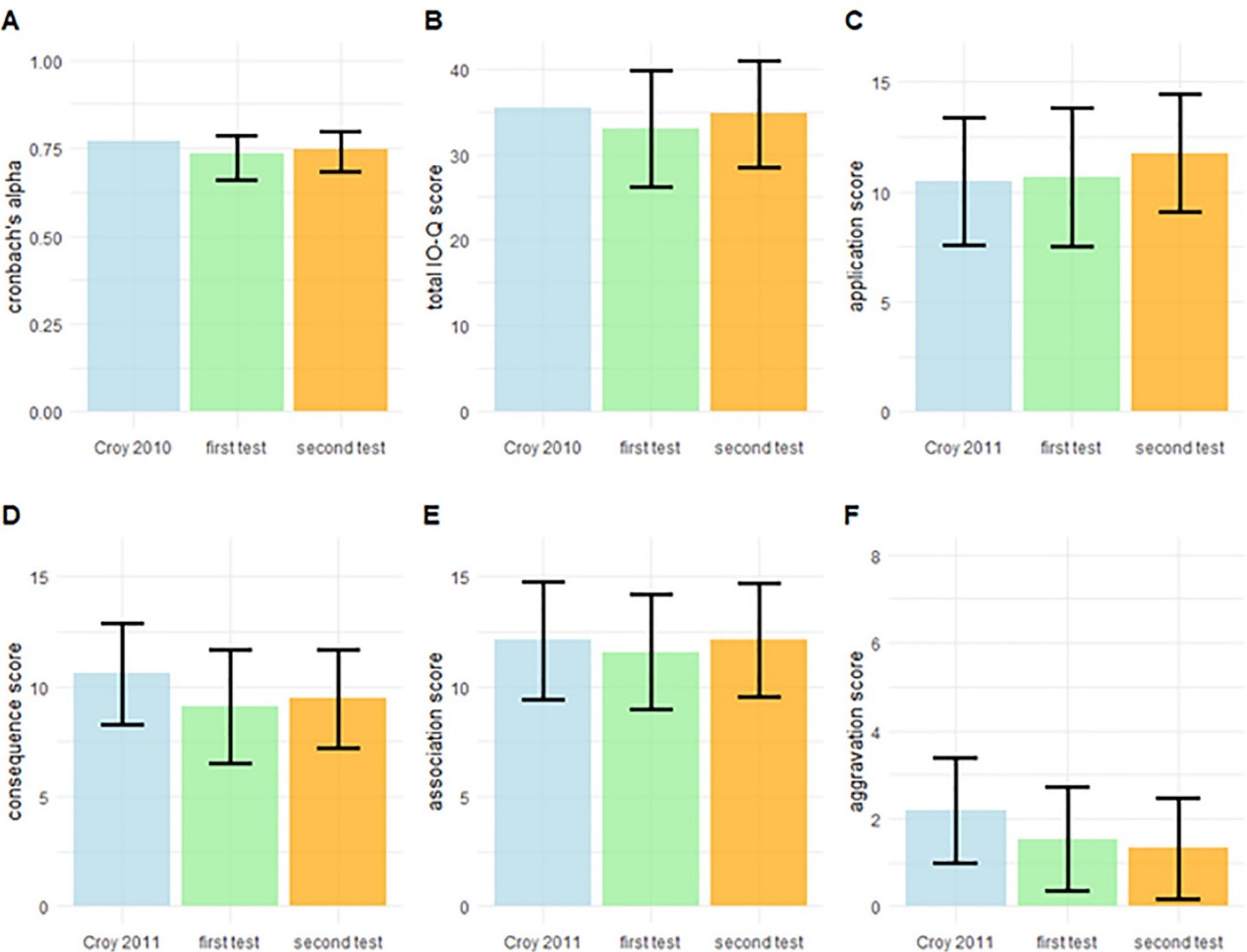

**Fig 2. Internal consistency and comparison between Tests 1 and 2.** A) Bar-charts showing (A) Cronbach's alpha for the IO-Q (error bars represent 95% confidence intervals) for tests 1 and 2 along with the value reported by Croy et al. (2010). B-F) Mean scores for tests 1 and 2 for total IOQ and subscales including the means reported by Croy et al. (2011). Error bars in B-F represent the standard deviations from the mean.

subscale ranged between 0.49 (Agg at time 1) and 0.79 (Ass at time 2). Correlations internally between subscales were generally weaker (0.21 to 0.43). For comparison, Fig 3F shows the corresponding values in a previous publication from Croy et al. in 2010 which are qualitatively highly similar. In order to examine the quantitative similarity between these three matrices, we

**Table 2. Bayesian paired t-test.**

| Measures | BF10 | Change in mean |
|---|---|---|
| first and second IO-Q | 77260.743 | 1.782 |
| first and second association | 37.828 | 0.547 |
| first and second application | 1247000.000 | 1.106 |
| first and second consequence | 0.985 | 0.335 |
| first and second aggravation | 7.036 | -0.207 |

The table shows the difference in test scores between tests 1 and 2. '*Change in mean*' denotes the change in mean from testing round 1 to testing round 2.

**Table 3. Bayesian two-sample t-test.**

| Measure | BF10 | Change from Croy 2011 |
|---|---|---|
| first association | 2.309 | -0.51 |
| second association | 0.085 | 0.04 |
| first application | 0.110 | 0.18 |
| second application | 8.687e+6 | 1.18 |
| first consequence | 6.707e+9 | -1.48 |
| second consequence | 4.681e+7 | -1.14 |
| first aggravation | 2.666e+9 | -0.66 |
| second aggravation | 1.866e+16 | -0.87 |

The table shows the difference between the results of Croy and colleagues (2011) and tests 1 and 2 respectively. 'Change from Croy 2011' denotes the positive or negative change in mean from the reported values by Croy and colleagues.

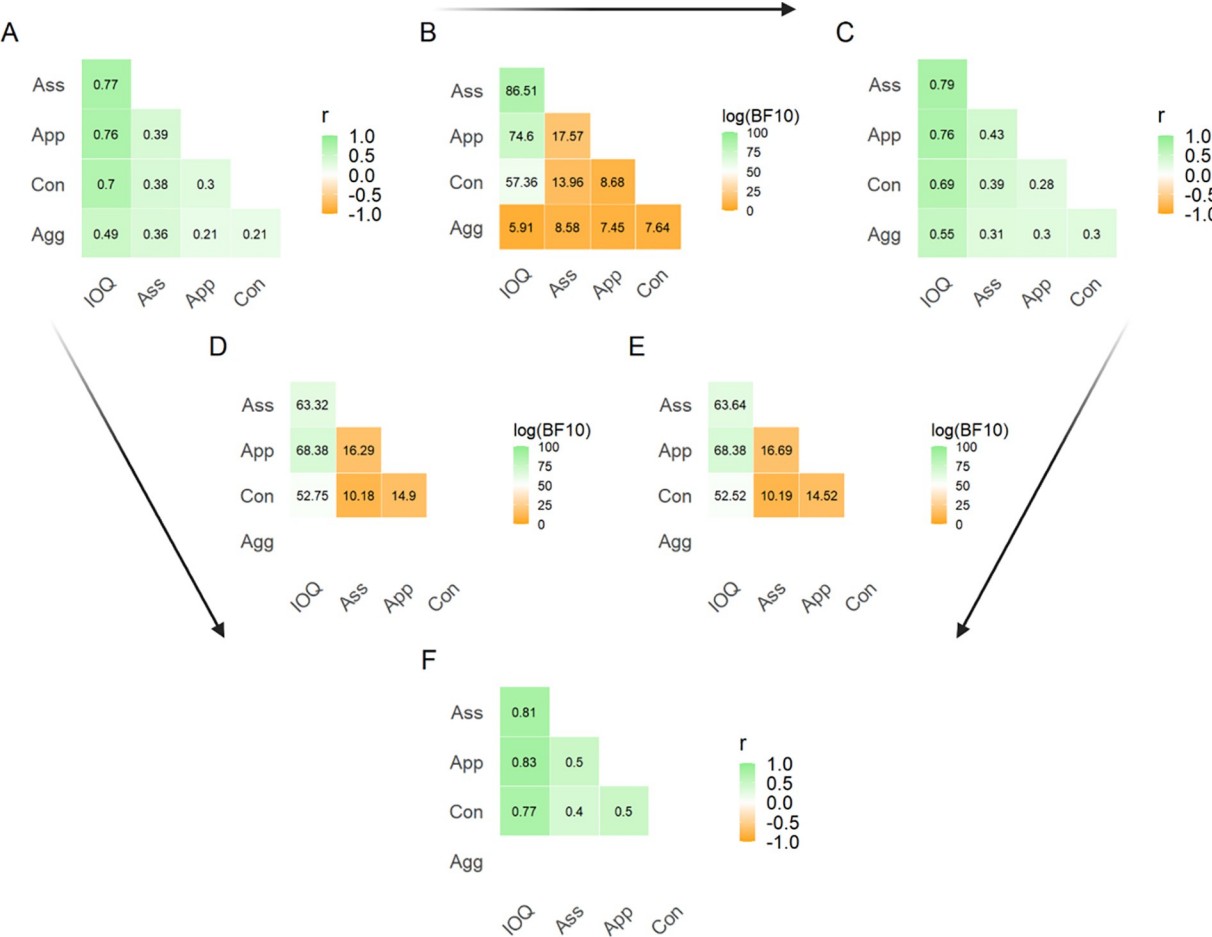

**Fig 3. Correlation matrices for total IO-Q and subscale scores.** A and C: Internal correlation matrices for test 1 (A) and test 2 (C) using Bayesian statistics with a uniform non-directional prior. F: Correlation matrix for values reported by Croy et al. (2010) for comparison (they did not report the Agg subscale). B, D, and E: Matrices reporting log(BF10) for replicative correlational analysis. Arrows point toward the matrix used as a replicative correlation and away from the matrix used as a prior. A value in this matrix of 0 would correspond to equal odds of the null hypothesis versus the alternate hypothesis being true. Note they all results have log(BF10) > 4.6, corresponding to BF > 100, i.e., extreme evidence for replication.

conducted three sets of Bayesian pairwise replication tests where one set of results (e.g., the test 1 results) was used as priors for another set of results (e.g., the test 2 results) and the resulting BF thus reflected the evidence in support of the first (test 1) results. This was done for all comparisons between test 1, test 2 and the results of Croy et al. in 2010. and the results are plotted in Fig 3B, 3D and 3E. Note that for all comparisons, log(BF) was greater than 4.6, meaning that there was extreme evidence in favor of the original effect in all cases. In sum, the internal correlations between the total IO-Q and subscale scores were nearly identical at our two tests despite the differences in language and format (and despite minor differences in the means), and the results reflected those previously reported for a German sample of 123 participants (mean age 36, range 12–68) [6].

### Test-retest reliability and influence of time

The test-retest reliability between tests 1 and 2 was examined using Pearson product-moment correlations. For total IO-Q, *r* was 0.78 (95% CI = [0.713; 0.830]) (Fig 4), while *r* was 0.69, 0.66, 0.67 and 0.70 for Ass, App, Con, and Agg, respectively (Table 4). The results of non-parametric analyses are reported in S1 Fig and S2 Table. Overall, test-retest reliability examined by

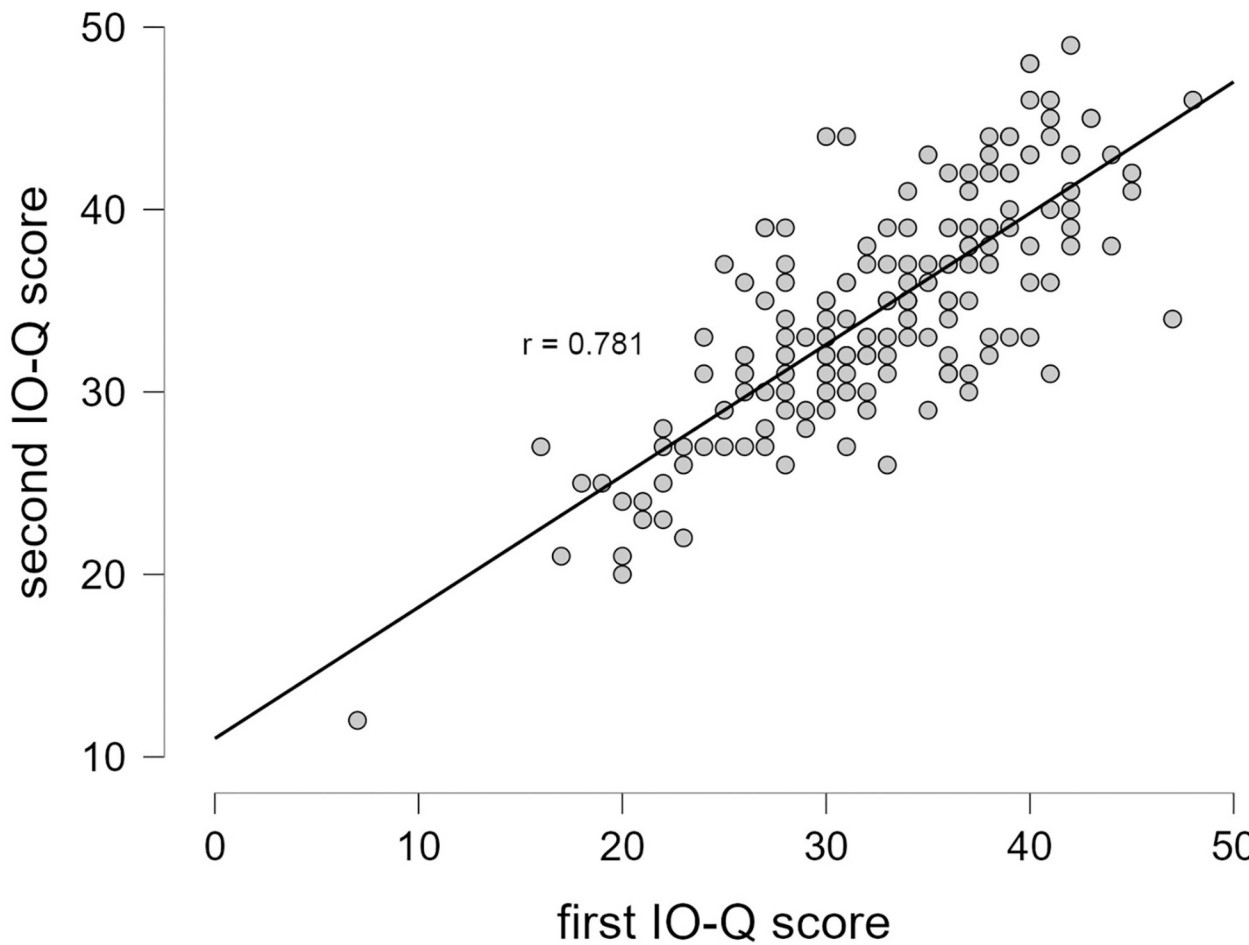

**Fig 4. Test-retest reliability.** Summarizing table (top) and a scatterplot (bottom) showing Bayesian Pearson correlation between the total IO-Q in tests 1 and 2. Dots may overlap.

**Table 4. Bayesian correlation for subscales.**

| Bayesian correlation | N | Pearson's r | BF10 | Lower 95% CI | Upper 95% CI |
|---|---|---|---|---|---|
| first IO-Q–second IO-Q | 179 | 0.781*** | 5.729e+34 | 0.713 | 0.830 |
| first ass–second ass | 179 | 0.688*** | 2.042e+23 | 0.598 | 0.755 |
| first app–second app | 179 | 0.660*** | 4.764e+20 | 0.564 | 0.733 |
| first con–second con | 179 | 0.668*** | 2.738e+21 | 0.574 | 0.739 |
| first agg–second agg | 179 | 0.696*** | 1.362e+24 | 0.608 | 0.762 |

Table presenting correlations between tests 1 and 2 for all subscales. app: Application, con: consequence, ass: association, and agg: aggravation.

* $BF_{10} > 10$,

** $BF_{10} > 30$,

*** $BF_{10} > 100$.

Pearson correlations could thus be described as acceptable, with the reliability of the subscales being slightly lower than that of the full IO-Q score. The intraclass correlation coefficient was 0.86 (CI = [0.77;0.91] (obtained in an ICC(2,k) test per Shrout and Fleiss convention). This result is highly similar to what was observed in a previous study using an Italian translation in both the first and the second test: 0.84 [18].

Fig 5 shows a histogram of time between tests 1 and 2 for all participants (Fig 5A) and a graphical posterior predictive check (Fig 5B). In order to examine the influence of time between tests on test-retest reliability, the difference between each participant's test score in tests 1 and 2 was calculated and normalized. Then, a linear model was fitted to the data (scaled difference in test score as a function of weeks between tests) and used to simulate data as replications ($y_{rep}$) of the outcome [19]. Kendall's tau for the correlation between the scaled difference and time between tests was 0.078 ($\tau = 0.078$, 95% CI = [-0.023;0.176], BF10 = 0.314), thus providing evidence for the null hypothesis that the difference in test scores did not vary over time. But since most participants completed the two tests within one month of each other, it is difficult to conclude beyond the range of weeks outside 0.5 to 6 weeks.

## Discussion

We translated the Importance of Olfaction Questionnaire (IO-Q) from English to Danish and presented both versions to a sample of normosmic, healthy, young, Danish volunteers in order to examine the test-retest reliability under the potential influence of language/modality of presentation. We first found that the questionnaire exhibited good internal consistency both when presented in Danish and in English (Cronbach's alpha around 0.73–0.75) and that the obtained values were similar to what has been reported previously (0.77; Croy et al., 2010). If the internal correlation between the total IO-Q and its components observed in the first test round was used as priors for the second test round, all Bayes factors were above 100, providing extreme evidence for a replication of the pattern of internal correlations. This was also the case when comparing the results of each test round to results obtained previously in a German sample by Croy and colleagues (2010).

The mean scores and standard deviations for the total IO-Q and subscales were comparable across languages and to those reported in other studies using the scale in normosmic, German samples, but small differences of a few points were observed in both cases. The difference over time was similar across samples, indicating that scale reliability could be the cause of the differences although a slight difference in report bias/criterion as a consequence of repeated testing could also be a possibility.

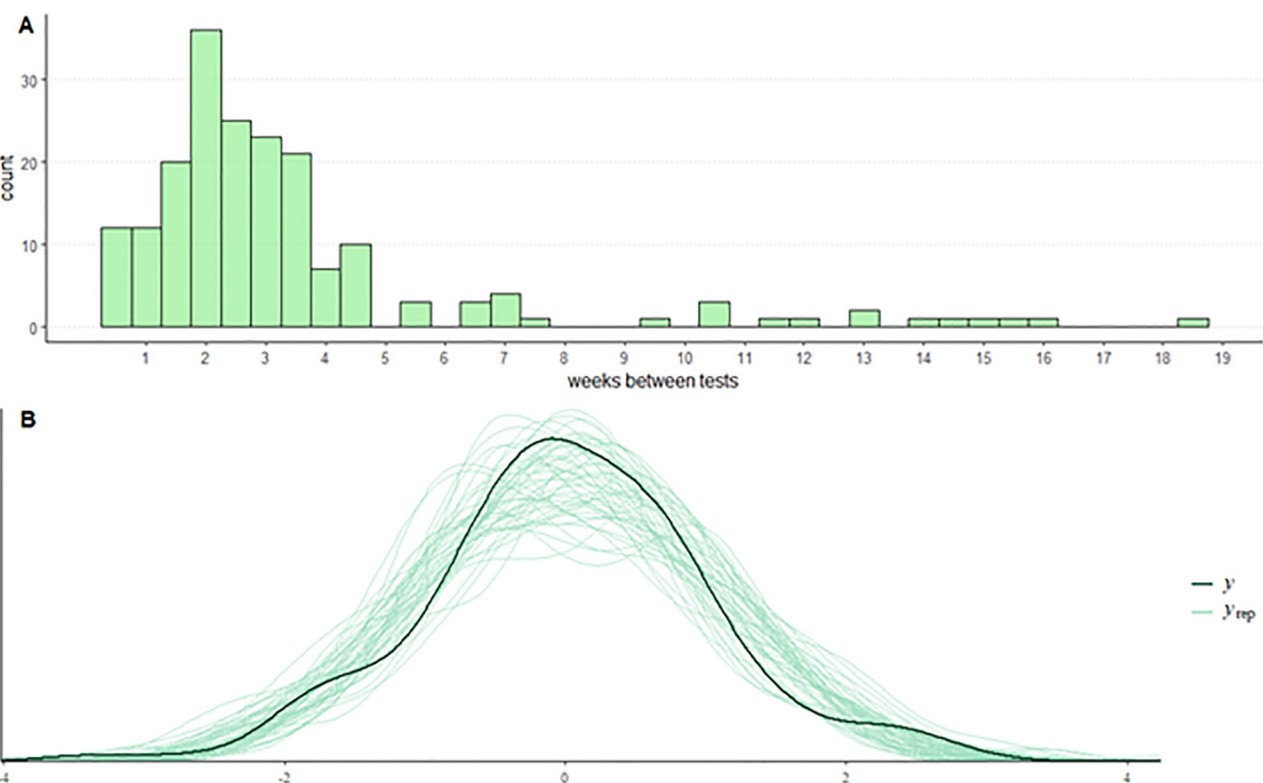

**Fig 5. Distribution of time between tests and a graphical posterior predictive check of the scaled difference between test score 1 and 2 versus time.** A) Histogram of the duration between tests 1 and 2. Bin width is 0.5 weeks. B) Using Bayesian statistics, a general linear model was fitted to the scaled difference between the total IO-Q score in tests 1 and 2 as a function of weeks between tests. If the general linear model is a good fit, then it should be able to produce data like currently observed data (y). Simulating an iterative run of 10,000 distributions (sample of those in light green, $y_{rep}$), it is approximately predicting this data and is centered around zero. Kendall's tau for the correlation between the scaled difference and time between tests was 0.078 ($\tau = 0.078$, 95% CI = [-0.023;0.176], $BF_{10} = 0.314$), indicating a minimal influence of time.

Taken together, the findings suggest that the influence of language of administration (Danish versus English) is minimal in a young, Danish sample and that normative values are relatively similar between Danish and German samples. This is important as it validates the usage of the Danish translation of the IO-Q, and because it allows easy and valid comparison between basic science studies often conducted with English-language questionnaires and clinical studies or practice typically using Danish-language questionnaires. Furthermore, the model for translation and validation may be applied in translations to other languages.

The test-retest reliability was examined by correlating the result obtained in test 1 (English) to that obtained in test 2 (Danish). For the total IO-Q, an acceptable test-retest correlation of 0.781 was obtained whereas the values were slightly lower for the subscales (range r: 0.63–0.70). Trecca et al., performed both the test and retest with a sample of 50 participants from a test population with an average age of 35.4 in Italian, digitally [18]. Applying the intraclass coefficient with two-way random, average measures and absolute agreement, they found an overall intraclass correlation coefficient of 0.84. We obtained a highly similar result of 0.86. We found no evidence for an influence of time on the test-retest variability although it is worth noting that it is difficult to conclude beyond durations of 5–6 weeks (for ≈84%, the duration was below 5 weeks).

## Limitations

One potentially confounding factor was that the first test was administered digitally in English whereas the second test was administered as a pen-and-paper test in Danish. This could have made it difficult to separate the effects of administration mode and language. However, since the means, distributions, internal consistency and subscale interrelationships were very similar for the two tests, it seems unlikely that either factor had more than an at most minimal influence on the results. Furthermore, given the good test-retest reliability, it seems likely that the influence of language and administration mode should be thought of mainly as a small difference in bias or random variation in study population.

One difference was that the pen-and-paper test format allowed participants to disobey instructions by, for example, reporting between response categories (marking the paper between boxes when in doubt between two response options) or not answering every question. To avoid any interpolation of the data whatsoever, these two features caused exclusion and, consequently, data loss.

## Conclusion

Despite differences in language and mode of administration, the IO-Q yielded highly similar data patterns across repeated testing, and test-retest reliability was good. The stability of the results obtained with the Danish versus English IO-Q validates its usage in Danish and its utility as a measure in clinical practice to evaluate patients' subjective experience of olfaction in general–both as digital and analogue application.

## Supporting information

**S1 Table. The individual Importance of Olfaction Questionnaire (IO-Q) in Danish.** The IO-Q has 20-items and reports are made on a 4-point Likert scale that is scored 0 for complete disagreement and 3 for complete agreement. The items are separable into four categories (association (Ass), consequence (Con), application (App), and aggravation (Agg)). Each of the subscales are made of 6 items except Agg with only 2. The Scale column is not printed when the questionnaire is presented to participants.
(DOCX)

**S2 Table. Comparison with other studies.** List of results in other studies [9, 20, 21]. Note that these studies differ either in how the IOQ was scored (IOQ18 using only the total App, Ass and Con while leaving out Agg) or in terms of the culture of the studies population (Asian or American rather than European).
(DOCX)

**S1 Fig. Correlation between first and second IOQ.** Non-parametric correlational scatterplot between the total score of the first test (Ranks of first IO-Q) and the second test (Ranks of second IO-Q). Notice the extreme evidence for positive correlation. Note that data points may overlap.
(TIFF)

## Acknowledgments

We thank Amalie Holm Lund Sørensen, Anna Villaume Stuckert, Bianka Szöllösi, Camile Maria Costa Correa, Dora Veraszto, Dunja Paunovic, Katarina Vulic, Magnus Knudsen, Povilas Tarailis, and Sara Viuf for their assistance in data collection.

## Author Contributions

**Conceptualization:** Alexander Wieck Fjaeldstad, Kristian Sandberg.

**Data curation:** Alexander Wieck Fjaeldstad, Kristian Sandberg.

**Formal analysis:** Daniel Tchemerinsky Konieczny.

**Funding acquisition:** Alexander Wieck Fjaeldstad, Kristian Sandberg.

**Investigation:** Daniel Tchemerinsky Konieczny, Alexander Wieck Fjaeldstad, Kristian Sandberg.

**Methodology:** Daniel Tchemerinsky Konieczny, Alexander Wieck Fjaeldstad, Kristian Sandberg.

**Project administration:** Daniel Tchemerinsky Konieczny, Alexander Wieck Fjaeldstad, Kristian Sandberg.

**Software:** Daniel Tchemerinsky Konieczny.

**Supervision:** Alexander Wieck Fjaeldstad, Kristian Sandberg.

**Visualization:** Daniel Tchemerinsky Konieczny.

**Writing – original draft:** Daniel Tchemerinsky Konieczny.

**Writing – review & editing:** Daniel Tchemerinsky Konieczny, Alexander Wieck Fjaeldstad, Kristian Sandberg.

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
