## [Decision Letter · Decision Letter 0]

10 Oct 2022

PONE-D-22-14194Test-retest reliability and validity of the Importance of Olfaction questionaire in DenmarkPLOS ONE

Dear Dr. Tchemerinsky Konieczny,

Thank you for submitting your manuscript to PLOS ONE. After careful consideration, we feel that it has merit but does not fully meet PLOS ONE’s publication criteria as it currently stands. Therefore, we invite you to submit a revised version of the manuscript that addresses the points raised during the review process.

ACADEMIC EDITOR: Please also consider my comments and suggestions.==============================

We look forward to receiving your revised manuscript.

Kind regards,

Sorana D. Bolboacă, Ph.D., M.Sc., M.D.

Academic Editor

PLOS ONE

2. Thank you for providing your data sharing statement: "Data cannot be shared publicly because it is part of an ongoing study and thus considered unanonymised under Danish law" PLOS ONE has specific criteria regarding data-sharing and availability (https://journals.plos.org/plosone/s/data-availability). Specifically, these guidelines require that authors to make all data necessary to replicate their study’s findings publicly available without restriction at the time of publication. When specific legal or ethical restrictions prohibit public sharing of a data set, authors must indicate how others may obtain access to the data. To that effect, please clarify in your data availability statement whether the data for your study can be made available at time of publication, if accepted.

3. We noted in your submission details that a portion of your manuscript may have been presented or published elsewhere. Please clarify whether this [conference proceeding or publication] was peer-reviewed and formally published. If this work was previously peer-reviewed and published, in the cover letter please provide the reason that this work does not constitute dual publication and should be included in the current manuscript.

Reviewers' comments:

Reviewer's Responses to Questions

**Comments to the Author**

1. Is the manuscript technically sound, and do the data support the conclusions?

Reviewer #1: Yes

Reviewer #2: Yes

Reviewer #3: Yes

2. Has the statistical analysis been performed appropriately and rigorously? 

Reviewer #1: Yes

Reviewer #2: Yes

Reviewer #3: Yes

3. Have the authors made all data underlying the findings in their manuscript fully available?

Reviewer #1: Yes

Reviewer #2: No

Reviewer #3: Yes

4. Is the manuscript presented in an intelligible fashion and written in standard English?

Reviewer #1: Yes

Reviewer #2: Yes

Reviewer #3: Yes

5. Review Comments to the Author

Reviewer #1: Thanks for inviting me to review this manuscript. It’s good but there are some things to correct:

1. Introduction is too long… only 3 paragraphs are enough.

2. The section Method contains numerical information… these are results. You have to correct the entire section.

3. Why you didn’t use only the Danish questionnaires? And why you have a different time between the digital and analogue questionnaires?

4. It’s not clear for me why you compared your results only with Croy and colab.? It’s a little bit unusual to see such analysis… there are a lot of studies performed by Prof. Hummel and his team…

5. The Danish version of IO-Q seems to be reliable, but something like a meta-analysis to compare your results seems to be more appropriate.

Reviewer #2: The authors set out to test and retest the reliability and validity of the Danish translation of the previously developed importance of olfaction questionnaire (IO-Q). They present in detail the IO-Q and its component sub-scales.

The IO-Q was tested and retested on a consistent sample of n=179. However, the authors do not present the recruitment strategy, nor if respondents were compensated in any way for taking part in the study.

Importantly, researchers dropped the data collected from a respondent that did not follow the procedure, as well as outlier data from respondents who completed the analogue Danish questionnaire within a longer timespan (p.6).

Overall, the manuscript is technically sound, the data and tests results persuasively support the conclusions, which are appropriately drawn. In addition, the authors are explicit with regards to the limits of results (page 17).

Concerning question 2 (Q2 in the review form), the authors clearly spell out the reasons for using Bayesian tests in the statistical analyses they had performed, detailing their strengths, as compared to traditional ones, in the task at hand. The article denotes rigour both methodologically and in explaining each research stage completed to analyse the data.

Moreover, the authors specify the tests used in every stage of their analysis (Bayesian correlation analysis; Pearson product moment correlation; Cronbach’s alpha test; Bayesian t-test; graphical posterior predictive check for the goodness of fit of the proposed model; Breusch-Pagan test for linearity and homoscedasticity).

The article includes comparisons to previously published results on the development of the importance of olfaction questionnaire (IO-Q) and on the translation of the questionnaire in Italian.

Descriptives and results of each tests are presented in tables and graphs for each stage of the analysis (both testing rounds, as total scores on the IO-Q and separately for each sub-scale, comparisons with previously published results, scatter plots on test-retest reliability including correlation coefficient).

A particular strength of this analysis is the examination of the influence of time on between tests on test-retest reliability (results presented in Fig 5, page 15 and explained on page 16), asserting its limits as well (for approximately 85% of the sample the duration was below 5 weeks between test and retest).

In addition, the authors included supplementary evidence for further inspection in the Appendix. Adding the English version of the questionnaire would be welcome for interested readers (eg. validation of the IO-Q in other languages).

With regards to making the data fully available (Q3), the authors clearly stated upon the submission of the article that data could not be shared since it is part of an ongoing study (acted according to the Danish law).

The language in the article (Q4) is clear, correct, and unambiguous. There is just a minor formulation that needs revision on page 18:

“…obtained with the Danish versus English IO-Q underscores validates its usage in Danish and its utility…”

Reviewer #3: 1. Maybe it would be better to change the keywords to the words not found in the title anymore such as the four subscales of the importance of olfaction. These are the association, application, consequence, and aggravation.

2. To spell out acronyms like BA because even if this is obvious already its acronyms used in other countries are not the same, say AB (in the Philippines). Also, AWF was mentioned multiple times in the paper but the definition was not stated. It may mean African Wildlife Foundation, Albama Writers’ Forum, Analysis Work File, etc. It would be better to spell it out even the first time, the word was mentioned. The same with EU COST, refereeing to European Cooperation in Science and technology and all acronyms used in the whole manuscript.

3. The link given in the manuscript, https://www.josineverhagen.com/?page_id=76 prompted me this . Maybe it would be better to explain the content on the file that can be accessed in the given link so that even if it will not be placed anymore.

4. Is it possible to give more information about the script by Josine Verhagen? And can you also mention in the paper who is Josine Verhagen? Not all are familiar with her. Though it can be seen in the references that she was mentioned as one of the authors cited in the paper since she was given importance, can you also mention that she is a data scientist or her credibility in your paper?

5. Can you also include the English version of the questionnaire aside from the Danish-translated questionnaire?

6. PLOS authors have the option to publish the peer review history of their article (what does this mean?). If published, this will include your full peer review and any attached files.

Reviewer #1: No

Reviewer #2: No

Reviewer #3: No

---

## [Author Response · Author response to Decision Letter 0]

6 Dec 2022

(response letter, also attached)

Dear reviewers,

We thank you for your comments, which have improved the manuscript. Below, we address each of the in detail. Note also that this version of our manuscript also contains changes based on previous editorial comments. 

Sincerely on behalf of the authors,

Daniel Tchemerinsky Konieczny 

 

Reviewer 1

“Thanks for inviting me to review this manuscript. It’s good but there are some things to correct:”

“1. Introduction is too long… only 3 paragraphs are enough.”

Reply: Thank you the review. We have reduced the length of the introduction as requested, specifically shortening the first paragraph, merging it with the second, and shortening the final paragraph.

“2. The section Method contains numerical information… these are

results. You have to correct the entire section.”

Reply: In the current version, all numerical information in the Methods pertain to participant characteristics. While technically at odds with PLOS guidelines, they are typically reported in the Methods in psychological research papers and browsing recent PLOS One psychology articles, we see that nearly all follow this convention: (e.g., articles 10.1371/journal.pone.0242842, 10.1371/journal.pone.0270787, 10.1371/journal.pone.0257393, 10.1371/journal.pone.0250905, 10.1371/journal.pone.0266204, 10.1371/journal.pone.0171793). 

For this reason, we would prefer to retain current organization.

“3. Why didn’t you use only the Danish questionnaires? And why you have a different time between the digital and analogue questionnaires?”

Reply: As most Danes have good English language skills, testing is frequently conducted in English in Denmark, for example to be able to include non-native-speakers. Indeed, the two versions of the questionnaire were proposed to the paradigm package by two different researchers who tested in English and Danish respectively, thus creating the opportunity for this validation study to be conducted in addition. The advantage of having both a Danish and an English version is that we had the opportunity to confirm that test language at most has a minor influence in Denmark (or Scandinavia), which will likely be relevant to researchers in the future.

As mentioned in “Methods”, the current study is part of a larger project. This project has a total of 7 experimental test sessions. As mentioned in “Methods: Procedure”, the time between sessions was determined by the test time of the two sessions. Participants could book these relatively freely in order to maximise the completion rate, but they were instructed to aim to complete all sessions within 5-6 weeks. 

“4. It’s not clear for me why you compared your results only with Croy and colab.? It’s a little bit unusual to see such analysis… there are a lot of studies performed by Prof. Hummel and his team…”

Reply: Thank you for pointing this out. We initially used the studies of Croy and colleagues as these were the first publications using the scale, but we have now performed a more extensive search to identify studies that could be relevant to compare to. Please see the next reply for more details.

“5. The Danish version of IO-Q seems to be reliable, but something like a meta-analysis to compare your results seems to be more appropriate.”

Reply: The manuscript now includes a brief overview of studies that can be compared to our results in the Appendix, and they are referred to in the Results section. We aimed to summarise them in a meta-analysis and compare our findings to this, but unfortunately several issues became apparent in the process that complicated the matter and raised doubt about the validity of such a procedure. 

A key issue was that previous studies used different reporting standards for the IOQ. For example, some reported the full 20-item score while others reported just the 18-item score excluding the aggravation subscale. Furthermore, some studies only reported the subscales but not the total score or vice versa, while others left out the aggravation score completely. This means we could not conduct a joint analysis. The study populations also varied greatly, and it was difficult to argue that – for example – samples different greatly in terms of age (children or elderly versus our young sample) or culture (e.g., Chinese, Indian and American versus our Danish sample) would be comparable. For this reason, we decided to retain the comparisons from the previous version, but in addition insert a supplementary table with the results of the previous studies so that they might individually be compared to our findings.

Reviewer 2

“The authors set out to test and retest the reliability and validity of the Danish translation of the previously developed importance of olfaction questionnaire (IO-Q). They present in detail the IO-Q and its component sub-scales. The IO-Q was tested and retested on a consistent sample of n=179.”

“However, the authors do not present the recruitment strategy, nor if

respondents were compensated in any way for taking part in the study.

Importantly, researchers dropped the data collected from a respondent that did not follow the procedure, as well as outlier data from respondents who completed the analogue Danish questionnaire within a longer timespan (p.6).”

Reply: Thank you for pointing this out. The following information has been added to the manuscript: "[The participants] were recruited through the Center of Functionally Integrative Neuroscience (Aarhus University) participant database and through local advertisement, and they were compensated 120DKK per hour of participation. 

“Concerning question 2 (Q2 in the review form), the authors clearly spell out the reasons for using Bayesian tests in the statistical analyses they had performed, detailing their strengths, as compared to traditional ones, in the task at hand. The article denotes rigour both methodologically and in explaining each research stage completed to analyse the data. Moreover, the authors specify the tests used in every stage of their analysis (Bayesian correlation analysis; Pearson product moment correlation; Cronbach’s alpha test; Bayesian t-test; graphical posterior predictive check for the goodness of fit of the proposed model; Breusch-Pagan test for linearity and homoscedasticity). The article includes comparisons to previously published results on the development of the importance of olfaction questionnaire (IO-Q) and on the translation of the questionnaire in Italian.

Descriptives and results of each tests are presented in tables and graphs for each stage of the analysis (both testing rounds, as total scores on the IO-Q and separately for each sub-scale, comparisons with previously published results, scatter plots on test-retest reliability including correlation coefficient).”

“A particular strength of this analysis is the examination of the influence of time on between tests on test-retest reliability (results presented in Fig 5, page 15 and explained on page 16), asserting its limits as well (for approximately 85% of the sample the duration was below 5 weeks between test and retest).”

“In addition, the authors included supplementary evidence for further inspection in the Appendix. Adding the English version of the questionnaire would be welcome for interested readers (eg. Validation of the IO-Q in other languages).”

Reply: Unfortunately, due to copyright issues we are not able to include the original English IO-Q.

“With regards to making the data fully available (Q3), the authors clearly stated upon the submission of the article that data could not be shared since it is part of an ongoing study (acted according to the Danish law).”

“The language in the article (Q4) is clear, correct, and unambiguous. There is just a minor formulation that needs revision on page 18:

“…obtained with the Danish versus English IO-Q underscores validates its usage in Danish and its utility…”

Reply: Thank you for noticing this error. We have deleted the words “underscores” and “useful” (later in the sentence) so that it now makes sense.

Reviewer 3

“1. Maybe it would be better to change the keywords to the words not found in the title anymore such as the four subscales of the importance of olfaction. These are the association, application, consequence, and aggravation.” 

Reply: Thank you for the suggestion. We have revised accordingly.

“2. To spell out acronyms like BA because even if this is obvious already its acronyms used in other countries are not the same, say AB (in the Philippines). Also, AWF was mentioned multiple times in the paper but the definition was not stated. It may mean African Wildlife Foundation, Albama Writers’ Forum, Analysis Work File, etc. It would be better to spell it out even the first time, the word was mentioned. The same with EU COST, refereeing to European Cooperation in Science and technology and all acronyms used in the whole manuscript.”

Reply: We have now spelled out all abbreviations and acronyms that we were able to identify, including EU COST. We have replaced “BA degrees” with “bachelor’s degrees”. For AWK, we explicitly state that it is “author AWK” thus referring to Alexander Wieck Fjaeldstad on the author list as is common practice.

“3. The link given in the manuscript, https://www.josineverhagen.com/?page_id=76 prompted me this . Maybe it would be better to explain the content on the file that can be accessed in the given link so that even if it will not be placed anymore.”

And

“4. Is it possible to give more information about the script by Josine Verhagen? And can you also mention in the paper who is Josine Verhagen? Not all are familiar with her. Though it can be seen in the references that she was mentioned as one of the authors cited in the paper since she was given importance, can you also mention that she is a data scientist or her credibility in your paper?”

Reply: Thank you for drawing attention to this. We have now added additional information about the method and its origin. Specifically, we briefly explain the method and refer to two papers where the method has previously been published. Should the link therefore at one time become unavailable, the two papers provide adequate explanation of the methods and the script used. The new text reads as follows:

“Priors in the replication correlation analysis were set by using the data from the first testing round to obtain a posterior distribution, which was then applied as the alternative hypothesis in the analysis. The method is previously published (20, 21), and R code is made freely available by one of the authors (Verhagen) (currently available at https://www.josineverhagen.com/?page_id=76). We used this code as a template for our analysis.”

“5. Can you also include the English version of the questionnaire aside from the Danish-translated questionnaire?”

Reply: Unfortunately, due to copyright issues we are not able to include the original English IO-Q.

---

## [Decision Letter · Decision Letter 1]

2 Jan 2023

Test-retest reliability and validity of the Importance of Olfaction questionaire in Denmark

PONE-D-22-14194R1

Dear Dr. Tchemerinsky Konieczny,

We’re pleased to inform you that your manuscript has been judged scientifically suitable for publication and will be formally accepted for publication once it meets all outstanding technical requirements.

Kind regards,

Sorana D. Bolboacă, Ph.D., M.Sc., M.D.

Academic Editor

PLOS ONE

------------------

Reviewer's Responses to Questions

**Comments to the Author**

1. If the authors have adequately addressed your comments raised in a previous round of review and you feel that this manuscript is now acceptable for publication, you may indicate that here to bypass the “Comments to the Author” section, enter your conflict of interest statement in the “Confidential to Editor” section, and submit your "Accept" recommendation.

Reviewer #1: All comments have been addressed

Reviewer #2: All comments have been addressed

2. Is the manuscript technically sound, and do the data support the conclusions?

Reviewer #1: Yes

Reviewer #2: Yes

3. Has the statistical analysis been performed appropriately and rigorously? 

Reviewer #1: I Don't Know

Reviewer #2: Yes

4. Have the authors made all data underlying the findings in their manuscript fully available?

Reviewer #1: No

Reviewer #2: Yes

5. Is the manuscript presented in an intelligible fashion and written in standard English?

Reviewer #1: Yes

Reviewer #2: Yes

6. Review Comments to the Author

Reviewer #1: Thanks for the answers. From my point of view the manuscript looks much better and can be accepted in this form. For the questionnaire in English I think a link can be provided.

Reviewer #2: Authors denote careful attention to the comments of the reviewers by adequately addressing all of them.

7. PLOS authors have the option to publish the peer review history of their article (what does this mean?). If published, this will include your full peer review and any attached files.

Reviewer #1: No

Reviewer #2: No

---

## [Editor Report · Acceptance letter]

1 Feb 2023

PONE-D-22-14194R1 

Test-retest reliability and validity of the Importance of Olfaction questionnaire in Denmark 

Dear Dr. Tchemerinsky Konieczny:

I'm pleased to inform you that your manuscript has been deemed suitable for publication in PLOS ONE. Congratulations! Your manuscript is now with our production department. 

Kind regards, 

on behalf of

Professor Sorana D. Bolboacă 

Academic Editor

PLOS ONE